# Dynamic reorganisation of intermediate filaments coordinates early B-cell activation

Carlson Tsui[1], Paula Maldonado[1], Beatriz Montaner[1,3], Aldo Borroto[2], Balbino Alarcon[2], Andreas Bruckbauer[1,4], Nuria Martinez-Martin[1,2], Facundo D Batista[1,5]

**During B-cell activation, the dynamic reorganisation of the cytoskeleton is crucial for multiple cellular responses, such as receptor signalling, cell spreading, antigen internalisation, intracellular trafficking, and antigen presentation. However, the role of intermediate filaments (IFs), which represent a major component of the mammalian cytoskeleton, is not well defined. Here, by using multiple super-resolution microscopy techniques, including direct stochastic optical reconstruction microscopy, we show that IFs in B cells undergo drastic reorganisation immediately upon antigen stimulation and that this reorganisation requires actin and microtubules. Although the loss of vimentin in B cells did not impair B-cell development, receptor signalling, and differentiation, vimentin-deficient B cells exhibit altered positioning of antigen-containing and lysosomal associated membrane protein 1 (LAMP1+) compartments, implying that vimentin may play a role in the fine-tuning of intracellular trafficking. Indeed, vimentin-deficient B cells exhibit impaired antigen presentation and delayed antibody responses in vivo. Thus, our study presents a new perspective on the role of IFs in B-cell activation.**

## Introduction

B cells play a critical role in providing adaptive immunity against pathogenic infections through the generation of antigen-specific antibodies. However, naive B cells must undergo activation to acquire these effector functions. Typically, B-cell activation is initiated via the engagement of the B-cell receptor (BCR) by cognate antigen (Harwood and Batista, 2010). Cross-linking of the BCR induces receptor-mediated signalling that drives different cellular processes, including membrane remodelling, cytoskeleton reorganisation, and the uptake of the antigen (Harwood and Batista, 2010). Internalised antigen is then processed and presented to T cells in the context of MHC-II molecules, which allows presenting

B cells to receive co-stimulatory signal from the T cells, typically via direct interaction of CD40L:CD40 or secreted cytokines such as IL-4 (Elgueta et al, 2009). This signalling synergy triggers robust cell proliferation and drives the differentiation to plasma cells or memory B cells (Kurosaki et al, 2010).

Although B cells can capture soluble antigen, they predominantly see antigen on the membrane of other APCs such as subcapsular sinus macrophages in vivo (Carrasco & Batista, 2007; Gaya et al, 2015). To gather and capture membrane-bound antigen from the APCs, B cells must alter their morphology and undergo spreading on the APCs (Fleire et al, 2006). Such realisation has since brought fresh attention to the role of cytoskeleton in B cells. Indeed, BCR signalling triggers rapid inactivation of the ezrin–radixin–moesin membrane linker and the release of the cortical actin cytoskeleton (Hao and August, 2005; Treanor et al, 2011). This allows B cells to rearrange their morphology and to accommodate the concurrent actin polymerisation to propagate the spreading response. Accordingly, depletion of the actin regulator Cdc42 or Rac2 renders B-cell spreading defective (Arana et al, 2008; Burbage et al, 2015). Moreover, loss of adaptor proteins of the actin cytoskeleton, such as Nck or WASP interacting protein, also alters the behaviour of B-cell spreading response (Castello et al, 2013; Keppler et al, 2015).

BCR stimulation also promotes rearrangement of the microtubule network. Indeed, the formation of an immunological synapse is associated with the rapid translocation of the microtubule organising centre (MTOC). This is thought to facilitate the trafficking of intracellular membrane compartments, such as lysosomes and TLR-9+ vesicles (Chaturvedi et al, 2008; Yuseff et al, 2011). Microtubule is also responsible for the trafficking of antigen after internalisation (Chaturvedi et al, 2008). Although MTOC translocation and targeted trafficking of lysosomes are thought to be important to release tightly bound antigens from stiff lipid surfaces (Yuseff et al, 2011; Spillane & Tolar, 2017), correct trafficking and positioning of antigen compartments are necessary to facilitate synergistic signalling and antigen presentation (Siemasko et al, 1998; Chaturvedi et al, 2008).

[1]Lymphocyte Interaction Laboratory, The Francis Crick Institute, London, UK  [2]Centro de Biología Molecular Severo Ochoa, Consejo Superior de Investigaciones Científicas, Universidad Autónoma de Madrid, Madrid, Spain  [3]Haematopoietic Stem Cell Laboratory, The Francis Crick Institute, London, UK  [4]Facility for Imaging by Light Microscopy, Imperial College London, London, UK  [5]Ragon Institute of Massachusetts General Hospital, Massachusetts Institute of Technology and Harvard, Cambridge, MA, USA

Correspondence: fbatista1@mgh.harvard.edu

Type III intermediate filament (IF) protein vimentin is a member of cytoskeleton networks highly expressed in B cells (Dellagi et al, 1982). Individual vimentin units assemble to form large filamentous bundles through multiple orders of dimerisation. Similar to f-actin or microtubule, vimentin filaments also undergo assembly and disassembly in a dynamic fashion (Goldman et al, 2008). In lymphocytes, its expression and filamentous distribution are associated with increased morphological stiffness of the cell (Brown et al, 2001). Accordingly, disruption of vimentin organisation renders the cells more prone to mechanical deformation. In line with this, vimentin-deficient lymphocytes cannot undergo extravasation via the trans-endothelial mechanism (Nieminen et al, 2006). Interestingly, it was also demonstrated that vimentin undergoes rapid reorganisation upon surface BCR cross-linking (Dellagi & Brouet, 1982). However, whether such dynamics or plasticity of vimentin plays a role in B-cell activation is unknown.

Here, using super-resolution imaging techniques, we show that the rapid collapse and reorganisation of the vimentin cytoskeleton is a general feature of BCR signalling, and it correlates with the intracellular trafficking of antigen and lysosomal associated membrane protein 1 (LAMP1$^+$) compartments. By characterising the vimentin-null mice, we show that vimentin is required to mediate intracellular trafficking and antigen presentation in B cells. We show that B cells lacking vimentin exhibit altered positioning of antigen and LAMP1$^+$ compartments, as well as reduced presentation capacity in the context of low antigen availability. Moreover, loss of vimentin in the B-cell compartment delayed antibody response and affinity maturation in vivo. Taken together, this study provides new insights into the function of IFs in B cells.

# Results

## BCR engagement induces the collapse and reorganisation of the vimentin cytoskeleton

Although it has been previously reported that B-cell activation results in the rapid formation of the vimentin cap (Dellagi & Brouet, 1982), it is not known how this is coordinated or how this is related to B-cell function. To further investigate into this phenomenon, we first compared the organisation of vimentin in primary B cells before and after BCR engagement by soluble antigens (Fig 1A–C). To this end, we compared vimentin distribution in resting B cells and B cells stimulated with F(ab)₂ anti-IgM using conventional confocal microscopy and Airyscan confocal super-resolution microscopy (Figs 1A and S1A). In most of the resting B cells, we observed a loose cage-like vimentin structure (Figs 1A and B, and S1A). However, we found that activated B cells exhibited a collapse and polarised vimentin structure (Figs 1A and B, and S1A). We confirmed the distribution of vimentin as cage-like structures in resting follicular B cells by imaging frozen splenic sections using Airyscan confocal microscopy (Fig S1B). To reveal greater details in the changes in the vimentin network after BCR cross-linking, we used structured illumination microscopy (SIM) to analyse the samples. In resting B cells, we observed an interconnected cage-like vimentin network that distributed uniformly around the cell with visible filamentous

elements (Fig 1C). As described, activated B cells exhibited collapsed and polarised vimentin distribution (Fig 1C). We observed filamentous elements at the periphery of the vimentin cluster; however, these were largely undefined in the centre of the cluster (Fig 1C), possibly because of the high density of vimentin protein present.

A cage-like vimentin distribution maintains the membrane stiffness of spherical lymphocytes (Brown et al, 2001). However, BCR cross-linking induces cell spreading that involves a dramatic reorganisation of the cell morphology and thus requires membrane flexibility (Fleire et al, 2006). To test whether vimentin collapse correlated with B-cell spreading, we investigated the reorganisation of vimentin in B cells stimulated using plate-bound antigen. To this end, we fixed B cells onto either poly-lysine–coated or anti-κ–coated coverslips. We then labelled the cells for vimentin and analysed them using confocal microscopy (Figs 1D and S1C). Consistent with our previous observations, B cells settled on poly-lysine remained spherical and exhibited a uniform cage-like vimentin network (Fig 1D). On the contrary, B cells undergoing spreading response on anti-κ–coated coverslips displayed collapsed and polarised vimentin (Fig 1D). Furthermore, we observed small perinuclear vimentin clumps in most of the fully spread B cells (Fig S1C), suggesting the spreading response is associated with extensive collapse and reorganisation of vimentin (Figs 1D and S1C).

We wondered whether this translocation of the cytoskeleton might involve structural changes of the filamentous vimentin within the cluster. To address this, we specifically imaged vimentin filaments at the spreading interface in super-resolution by combining direct stochastic optical reconstruction microscopy (dSTORM) and total internal reflection microscopy (TIRF) (Fig 1E). This approach enabled lateral resolution up to 10 nm, thus allowing single vimentin filaments to be resolved (Xu et al, 2012). This technique also limited the depth of illumination of the specimen to 100 nm, and thus only imaged events close to the plasma membrane. We found that whereas resting B cells exhibited loose filaments of vimentin within the penetrable depth of the interface (Fig 1E), B cells settled on anti-κ displayed extensive vimentin recruitment towards the interface (Fig 1E). Notably, we observed little apparent changes in the integrity or thickness of the filaments, indicating that the recruitment of vimentin to the interface potentially involved little structural changes of the cytoskeleton. Similar to the observations using confocal microscopy, in fully spread B cells that exhibited the typical ring-like actin lamellae and dot-like actin foci, we observed that the vimentin filaments underwent extensive polarisation and withdrew from the lamellae of the cells (Fig 1E). In addition, the centre of the vimentin clump became extremely dense and possibly represented overlapping filaments (Fig 1E). Nonetheless, we did observe filamentous elements at the periphery of the clump (Fig 1E).

In vivo, BCR engagement by membrane-bound antigen on APCs triggers the formation of the immunological synapse. To see whether antigen-engaged BCR would recruit and maintain vimentin at the immunological synapse, we next visualised the distribution of vimentin in B cells stimulated with anti-IgM–coated large microspheres (∅ 4.95 μm) using confocal microscopy (see the Materials and Methods section) (Fig 1F and G). In line with our predictions, we observed that although cells not in close contact

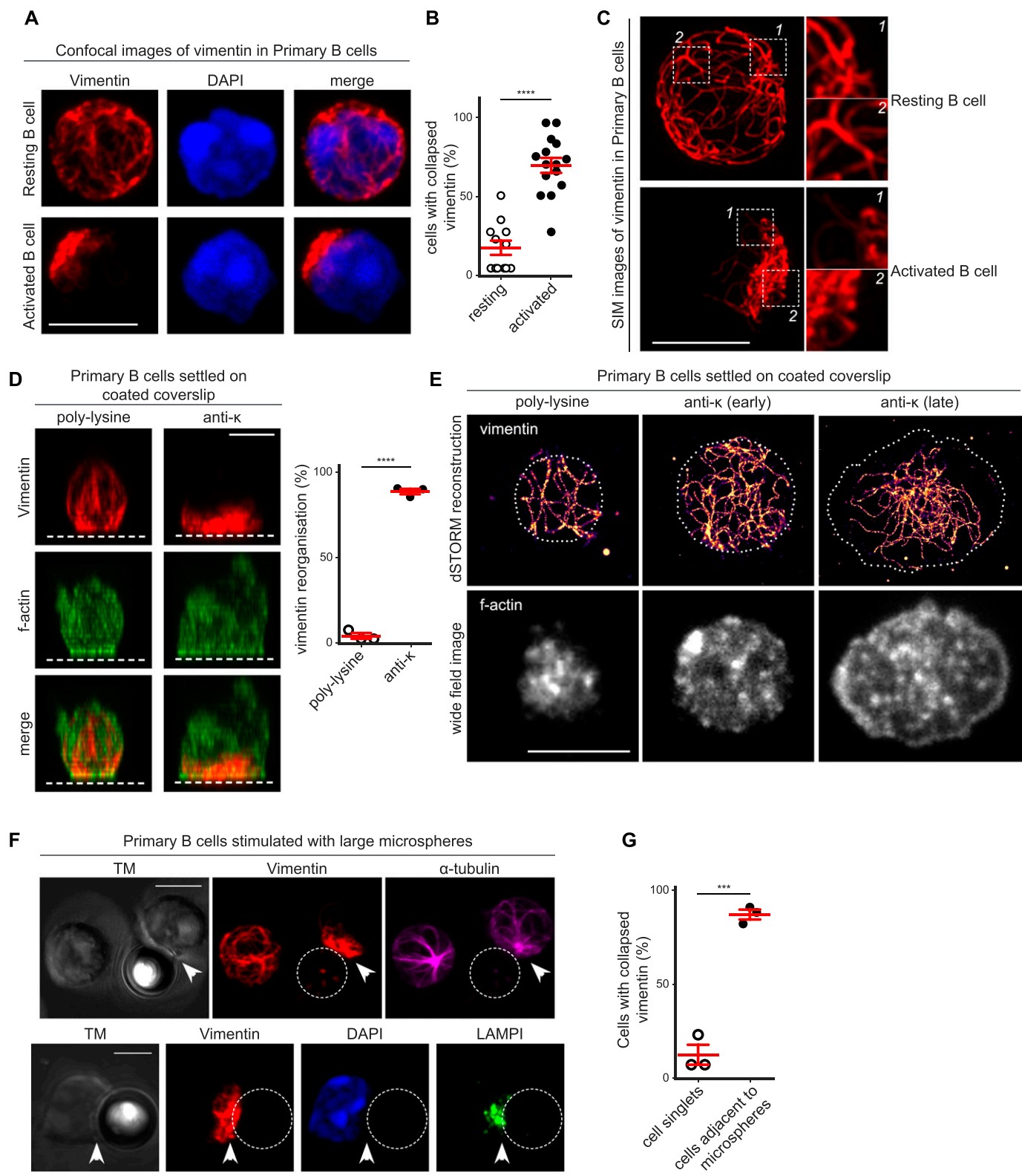

**Figure 1. Rapid vimentin reorganisation upon BCR stimulation.**
**(A–C)** Resting primary B cells and B cells stimulated with 5 μg/ml anti-IgM F(ab)₂ fragment for 15 min were settled onto poly-lysine–coated coverslips and fixed. The cells were labelled for vimentin and processed for (A, B) confocal microscopy or (C) SIM. **(A)** Representative confocal image showing the distribution of vimentin in resting (A, upper panel) and activated (A, lower panel) B cells. Scale bar = 5 μm. **(B)** The degree of vimentin collapse was quantitated. **(C)** Representative images and magnifications of the structure of vimentin obtained using SIM. Scale bar = 5 μm. **(D)** Primary B cells were settled onto poly-lysine–coated or anti-κ–coated

with microspheres displayed uniform cage-like vimentin (Fig 1F and G), almost all of the cells that were adjacent to the microspheres exhibited collapsed and polarised vimentin towards them (Fig 1F and G), suggesting that vimentin is specifically targeted to the interface. Importantly, these cells also displayed translocated MTOC (Fig 1F) and LAMP1$^+$ compartments (Fig 1F), suggesting that these conjugates were genuine (Yuseff et al, 2011). Taken together, these results suggest that the selective reorganisation of vimentin is targeted to antigen-engaged BCRs and might play a role in facilitating morphological changes in B cells during the formation of an immunological synapse.

### Vimentin collapse and polarisation is associated with intracellular trafficking of antigen and LAMP1$^+$ compartments

BCR engagement by soluble antigen initiates the internalisation and subsequent intracellular trafficking of the BCR–antigen complex. Internalised BCR–antigen remains signalling-competent and continues to recruit signalling molecules as it undergoes trafficking and polarises into a perinuclear cluster associating with the MTOC (Siemasko et al, 1998; Chaturvedi et al, 2008; Tsui et al, 2018). To extend our characterisation of the reorganisation of vimentin in this context, we visualised the distribution of vimentin in cells after antigen internalisation. To this end, we initially stimulated primary B cells using Alexa647-conjugated F(ab)$_2$ anti-IgM for 30 min. We then fixed and imaged the samples using SIM (Fig 2A and B). We observed that the polarisation of vimentin strongly correlated with the polarisation of antigen (Fig 2A and B), suggesting that vimentin collapse and polarisation might be involved in the intracellular trafficking of antigen. To better dissect the spatial relationship between vimentin and intracellular antigen, we analysed single z-stacks of the SIM images to characterise the localisation of both vimentin and antigen in these cells (Fig 2C). To our surprise, despite the apparent correlation of polarisation in both compartments (Fig 2B), we observed a strong mutual exclusivity between both compartments (Fig 2C). Similar observations were made using the Airyscan super-resolution microscopy (Fig 2D). In addition, we observed a preferential residence of intracellular antigen in vimentin-void "pockets" that were encompassed by neighbouring vimentin filaments (Fig 2C and D). We noticed that this was independent of the size of the antigen-containing compartments, as we also observed the same trend with large ring-like, antigen-containing structures (Fig 2D) (Martínez-Martín et al, 2017). To see whether the reorganisation of vimentin also involved the recruitment of other compartments in this context, we next visualised the localisation of LAMP1$^+$ compartments in primary B cells stimulated with anti-IgM (Styers et al, 2004; Chaturvedi et al, 2008; Yuseff et al, 2011). To do this, we imaged the distribution of vimentin and LAMP1 in activated B cells using confocal microscopy. Accordingly, we found that whereas

resting B cells exhibited nonpolar distribution of LAMP1$^+$ compartments, activated B cells exhibited polarised LAMP1$^+$ compartments that colocalised strongly with vimentin (Fig 2E), suggesting that vimentin might be involved in lysosome recruitment. Taken together, our data demonstrate that the trafficking of intracellular antigen and LAMP1$^+$ compartments are associated with the dynamic reorganisation of vimentin and that vimentin might play a role in the regulation of trafficking of these two intracellular compartments.

### Inhibition of early BCR signalling impairs vimentin reorganisation

Given that vimentin underwent reorganisation immediately after BCR stimulation, we wondered whether the propagation of BCR signalling was necessary to induce vimentin reorganisation in B cells. To address this, we first developed a high-throughput approach to characterise vimentin reorganisation in fixed primary B cells (see the Materials and Methods section). We characterised artificially generated "spots" of vimentin intensity to determine the status of the cytoskeleton (Fig 3A). Accordingly, mean spot intensity and area increase when vimentin is polarised, which we validated in activated B cells compared with resting B cells (Fig 3A and B). Next, we investigated whether pharmacological inhibition of Syk (which represented early BCR signalling), mitogen-activated protein kinase kinase (MEK), or mechanistic target of rapamycin complex (mTORC1) (which represented late BCR signalling) would affect vimentin reorganisation (Fig 3C). We found that inhibition of Syk significantly impeded vimentin reorganisation (Fig 3D). However, we did not observe any effects with B cells treated with MEK inhibitor or rapamycin (mTORC1 inhibitor) (Fig 3D), suggesting that the mTORC1 and MAPK signalling played little role in inducing vimentin reorganisation and that vimentin reorganisation was a consequence of early BCR signalling.

### Vimentin reorganisation requires functional actin and microtubule cytoskeleton

The steroidal lactone compound withaferin A (WFA) is known to induce covalent modification of vimentin (Bargagna-Mohan et al, 2007). We next studied the effects of WFA on vimentin organisation in resting and activated B cells. Using confocal microscopy, we observed that although resting B cells treated with WFA exhibited some observable aggregation of vimentin (Fig 4A) (Bargagna-Mohan et al, 2007; Grin et al, 2012), the nonpolar nature of vimentin distribution remained largely unaffected (Fig 4A). However, compared with control B cells, significantly fewer WFA-treated B cells underwent vimentin reorganisation in response to BCR stimulation (Fig 4B), suggesting that WFA inhibited BCR-induced vimentin reorganisation.

Despite its specific modification induced in vimentin, WFA also affects microtubules and actin (Grin et al, 2012). We wondered whether the inhibitory effects exerted by WFA on vimentin

coverslips and fixed. Intracellular vimentin and actin were labelled using anti-vimentin antibody and phalloidin, respectively. **(D)** Representative images showing lateral view of the cell from the 3D reconstruction of multiple z-stacks obtained using confocal microscopy. The white dotted lines depict the side of the coverslip. The degree of vimentin reorganisation was quantified for the two samples. **(E)** Representative corresponding dSTORM reconstruction of vimentin staining and the wide-field TIRF images of f-actin of cells settled on poly-lysine–coated and anti-κ–coated coverslips. **(F, G)** Primary B cells were stimulated with microspheres (∅ = 4.95 μm) coated with biotinylated anti-IgM. Excess microspheres were removed and the cells were settled onto poly-lysine–coated coverslips and fixed. **(F)** The cells were labelled for vimentin, tubulin, and LAMP1 and processed for confocal microscopy. Scale bar = 5 μm. **(G)** The distributions of vimentin in cells close to and distant from microspheres were quantitated. All data are representative of three independent experiments.

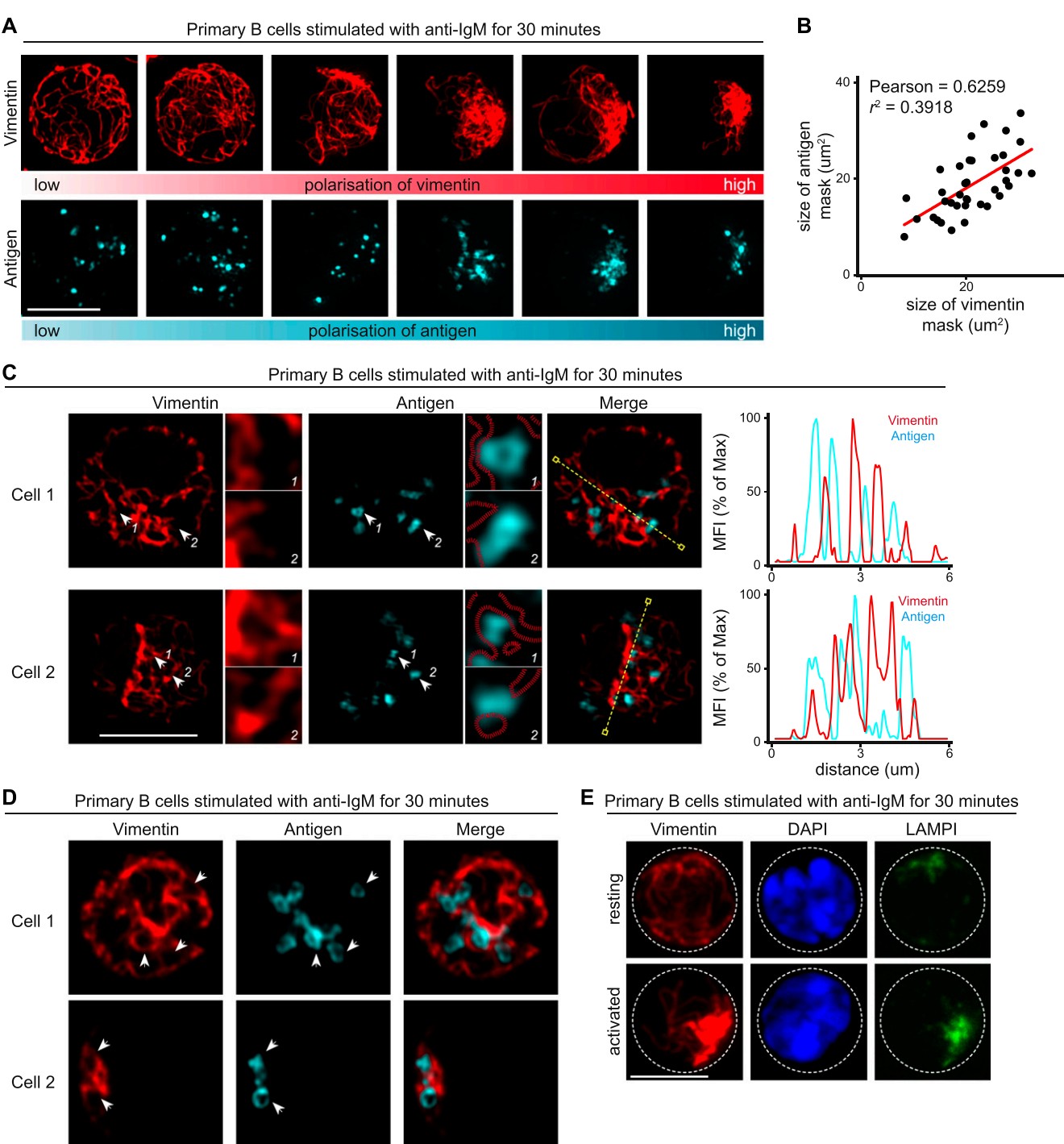

**Figure 2. Vimentin reorganisation correlates with intracellular antigen trafficking.**
**(A–C)** Primary B cells were stimulated with Alexa647-conjugated anti-IgM for 30 min at 37°C. The cells were then washed to remove excess antibody, and settled onto poly-lysine–coated coverslips and fixed. The cells were then permeabilised and labelled for vimentin. The samples were analysed using SIM. **(A)** Representative SIM images showing intracellular vimentin and the corresponding intracellular antigen. Scale bar = 5 $\mu$m. **(B)** The area coverage of vimentin intensity and antigen intensity was determined independently and correlation was determined. **(C)** Individual z-stacks from single cells were analysed and colocalisation of the two channels determined and depicted as histograms. Red mask (dashed lines) represents the area covered by vimentin. **(D)** Representative individual z-stacks from single cells imaged using Airyscan confocal microscopy showing the intracellular distribution of vimentin and antigen-containing compartments. **(E)** Representative confocal images showing the intracellular distribution of vimentin and LAMP1[+] compartments in resting (left panels) and activated (right panels) primary B cells. Resting primary B cells and B cells stimulated with anti-IgM F(ab)$_2$ for 15 min at 37°C were fixed onto poly-lysine–coated coverslips and labelled for vimentin and LAMP1. The samples were then processed for confocal microscopy. Scale bar = 5 $\mu$m. Data are representative of three independent experiments.

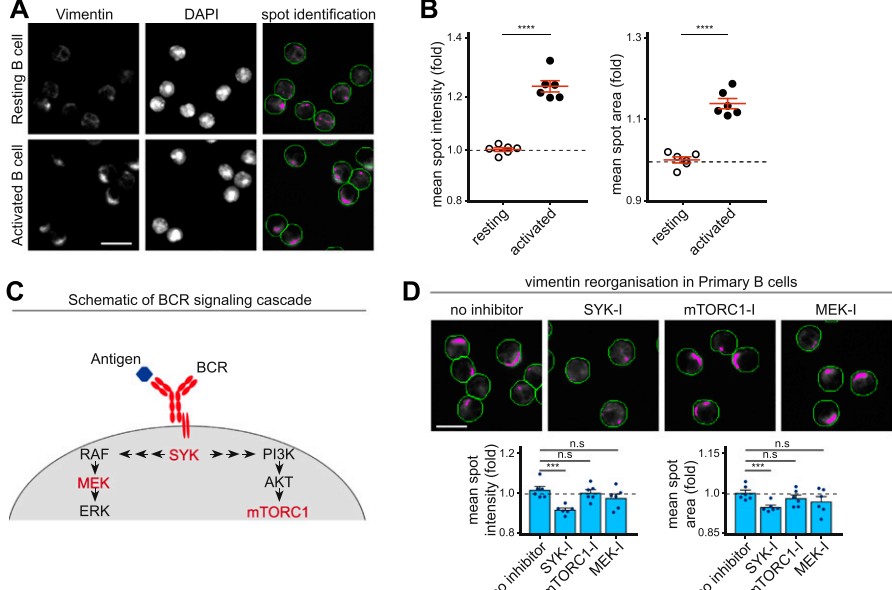

**Figure 3. Vimentin reorganisation requires early BCR signalling.**
**(A, B)** Primary B cells were stimulated using anti-IgM (F(ab)$_2$) for 15 min at 37°C. The cells were then settled onto poly-lysine–coated 96-well microplate and fixed. The cells were labelled for vimentin and then imaged by the Cellomics ArrayScan system. Data were analysed and spots were characterised using HSC navigator software. **(A)** Representative images and spot identification of resting and activated B cells. Scale bar = 10 μm. **(B)** The mean spot intensity (B, left panel) and the mean spot area (B, right panel) were quantitated. **(C, D)** Primary B cells pretreated with the specified inhibitors were stimulated using anti-IgM F(ab)$_2$ for 15 min at 37°C. The cells were then settled onto poly-lysine–coated 96-well microplate and fixed. The cells were labelled for vimentin, imaged by the Cellomics ArrayScan system and analysed for spot identification. **(C)** The schematic of BCR signalling cascade showing the targets of the inhibitors (in red). **(D)** The representative images and quantifications of mean spot intensities and mean spot areas. Scale bar = 10 μm. All experiments are representative of three independent experiments.

reorganisation involved the other two cytoskeleton systems, which are both known to interact with vimentin (Chang & Goldman, 2004). To establish this, we initially studied the effect of microtubule disruption on vimentin reorganisation using confocal microscopy. In line with previous observations (Brown et al, 2001), we found that microtubule depolymerisation did not induce major changes in the cage-like distribution of vimentin in resting B cells (Fig 4C), suggesting that the stability of vimentin structure in B cells did not require microtubules (Brown et al, 2001). Next, we compared the vimentin reorganisation in control B cells and nocodazole-treated B cells by antigen stimulation. We observed a significant reduction in vimentin reorganisation in nocodazole-treated B cells compared with control B cells (Fig 4D), suggesting that an intact and functional microtubule system was required in part for the reorganisation of vimentin upon BCR cross-linking (Yoon et al, 1998).

Next, we tested whether vimentin reorganisation required the actin cytoskeleton. To this end, we compared the distribution of vimentin in control B cells and B cells treated with latrunculin A (Lat A—an actin depolymerising agent). In B cells, Lat A treatment induces antigen-independent BCR-dependent signalling (Treanor et al, 2010; Mattila et al, 2013; Gasparrini et al, 2016). Interestingly, Lat A treatment in resting B cells did not cause vimentin reorganisation (Fig 4E), indicating that BCR signalling alone could not induce vimentin reorganisation. However, cells pretreated with Lat A exhibited impaired vimentin reorganisation in response to BCR cross-linking (Fig 4F), whereas the residence of antigen-containing compartments in vimentin-void pockets remained unaffected (Fig 4G). Our results suggest that the actin cytoskeleton was required for the reorganisation of vimentin (Yoon et al, 1998), which corroborates with our earlier findings on its dependence on early BCR signalling, given that rapid remodelling of the actin cytoskeleton is one of the consequences of early BCR signalling (Hao and August, 2005). Taken together, these results indicate that activation-induced reorganisation of vimentin depends on the dynamics of actin and microtubules.

Vimentin is known to interact and cooperate with the actin cytoskeleton in multiple scenarios (Esue et al, 2006; Schoumacher et al, 2010; Keeling et al, 2017). To validate our hypothesis using a genetic model, we analysed the dynamics of vimentin in $Cdc42^{-/-}$ B cells. Although active CDC42 can induce vimentin reorganisation (Meriane et al, 2000), CDC42 had been previously reported to play an important role in actin polymerisation, antigen polarisation, lysosome trafficking, and antigen presentation in B cells (Yuseff et al, 2011; Burbage et al, 2015). Thus, to see whether vimentin reorganisation would also be affected in $Cdc42^{-/-}$ B cells, we compared the distribution of vimentin, LAMP1$^+$, and antigen-containing compartments in activated WT and $Cdc42^{-/-}$ B cells using confocal microscopy. Consistent with our hypothesis, we found that around 70% of WT B cells exhibited collapse of vimentin (Fig 4H), whereas only 30% of $Cdc42^{-/-}$ B cells reorganised vimentin upon BCR cross-linking (Fig 4G). In line with previous findings, we observed that polarisation of both LAMP1$^+$ and antigen-containing compartments was also affected in $Cdc42^{-/-}$ B cells (Fig 4I) (Burbage et al, 2015), which correlated with reduced colocalisation between the two compartments (Fig 4J) (Chaturvedi et al, 2008).

## Vimentin is required for the fine-tuning of intracellular trafficking in B cells

We next asked whether the absence of vimentin affected B-cell activation, especially in the context of cell spreading, immunological synapse formation, and antigen polarisation. To this end, we functionally compared primary B cells from WT and vimentin-deficient mice ($Vim^{-/-}$) (Fig S2A and B) using in vitro approaches. Although specific functional differences were reported in vimentin-deficient lymphocytes, we observed no impairments in the development of the B-cell lineage both in the bone marrow and in the spleen in these animals (Fig S2C and D). To explore whether vimentin per se is required for early B-cell response when

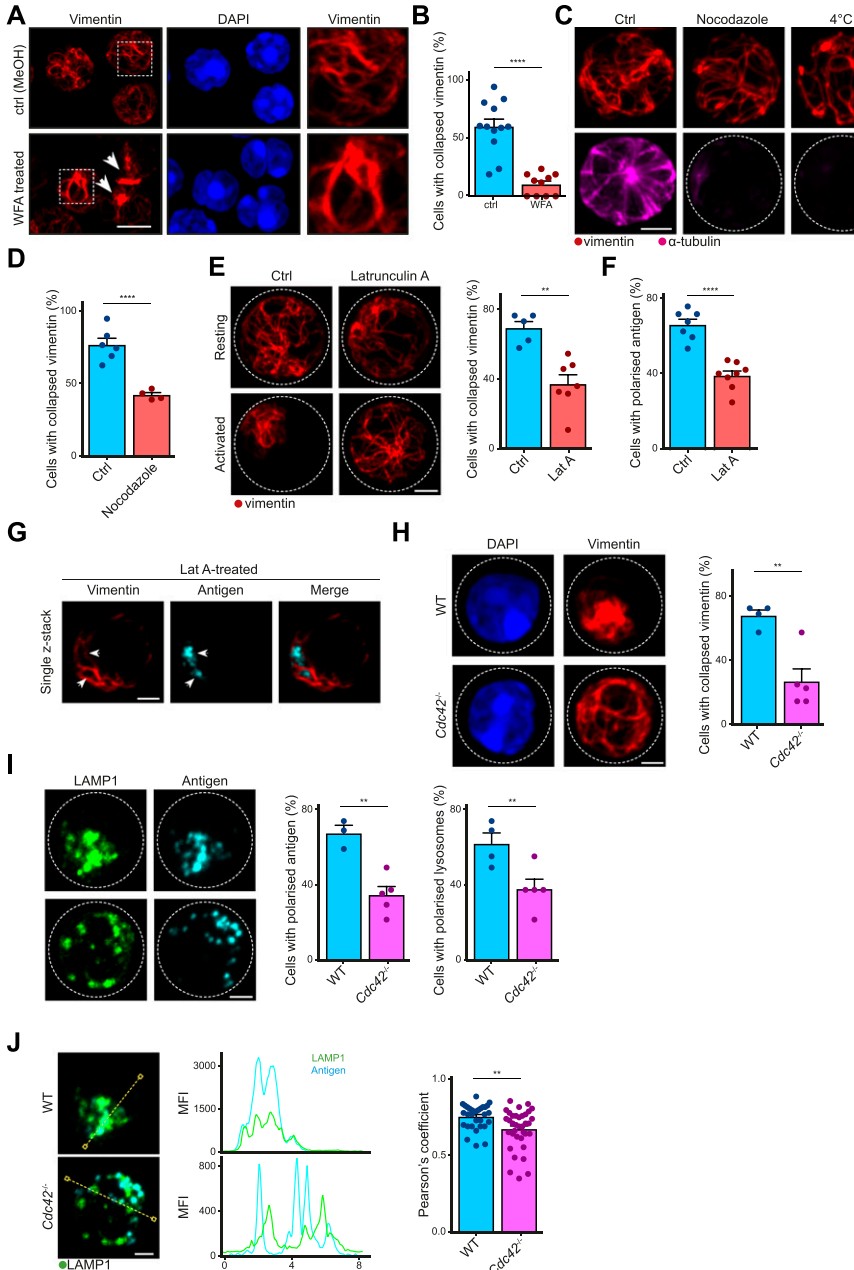

**Figure 4. Vimentin reorganisation requires actin and microtubules.**
**(A)** Representative confocal images showing the vimentin cytoskeleton in primary B cells in the absence (upper panel) and presence (lower panel) of WFA. Scale bar = 5 μm. **(B)** Quantification showing the degree of vimentin reorganisation in stimulated primary B cells treated with or without withaferin. **(C)** Representative confocal images showing distribution of the vimentin cytoskeleton in resting primary B cells where the microtubule cytoskeleton was perturbed. Primary B cells were purified and microtubule cytoskeleton was depolymerised using either nocodazole or incubation at low temperature. The cells were then fixed onto poly-lysine–coated coverslips and processed for confocal microscopy. Scale bar = 2 μm. Data are representative of three independent experiments. **(D)** Quantification of the degree of vimentin reorganisation in response to BCR stimulation in the control cells and in cells treated with nocodazole. t test was used to determine statistical significance. **(E)** Representative images of Airyscan confocal microscopy showing the distribution of intracellular vimentin in B cells treated with latrunculin A. Quantification shows the proportion of B cells with reorganised vimentin after BCR stimulation in the absence or presence of Lat A. Scale bar = 2 μm. **(F)** Quantification of the degree of antigen localisation in response to BCR stimulation in the control cells and in cells treated with Lat A. t test was used to determine statistical significance. **(G)** Representative single z-stack image of Airyscan confocal microscopy showing the distribution of intracellular vimentin and antigen in B cells treated with latrunculin A. Scale bar = 2 μm. **(H)** Representative confocal images and quantification showing vimentin distribution after BCR stimulation in WT and Cdc42−/− B cells. Scale bar = 2 μm. **(I)** Representative confocal images and quantification showing the distribution of antigen-containing and LAMP1+ compartments after BCR stimulation in WT and Cdc42−/− B cells. Scale bar = 2 μm. **(J)** Representative confocal images and analysis of colocalisation showing the distribution of antigen-containing and LAMP1+ compartments after BCR stimulation in WT and Cdc42−/− B cells. Scale bar = 2 μm.

stimulated with antigen, we initially compared the BCR signalling cascade and spreading response in WT and $Vim^{-/-}$ B cells upon BCR cross-linking. We observed no difference in both BCR signalling and the spreading response between WT and $Vim^{-/-}$ B cells (Fig S3A and B), suggesting that the absence of vimentin did not restrict the morphology-altering processes during B-cell activation. To confirm this, we next tested the ability of $Vim^{-/-}$ B cells in forming conjugates using confocal microscopy (Fig S3C and D). To this end, we incubated WT and $Vim^{-/-}$ B cells with anti-IgM–coated microspheres for 30 min and observed conjugates formation in the context of MTOC translocation. We found that both WT and $Vim^{-/-}$ B cells were able to form conjugates and exhibited correct MTOC

translocation (Fig S3C). Interestingly, we found that vimentin-deficient B cells were also able to recruit LAMP1+ compartments to the artificial immunological synapses (Fig S3D), suggesting that lysosome recruitment did not require vimentin. Similarly, we found no difference in the internalisation of antigen-engaged BCR between WT and $Vim^{-/-}$ B cells (Fig S3E). Collectively, our data indicated that lack of vimentin does not hinder morphology-altering processes during B-cell activation.

We wondered whether the loss of vimentin would affect intracellular trafficking of intracellular antigen and lysosomal compartments. To this end, we stimulated primary B cells using Alexa647 F(ab)₂ anti-IgM for 30 min and imaged the localisation of

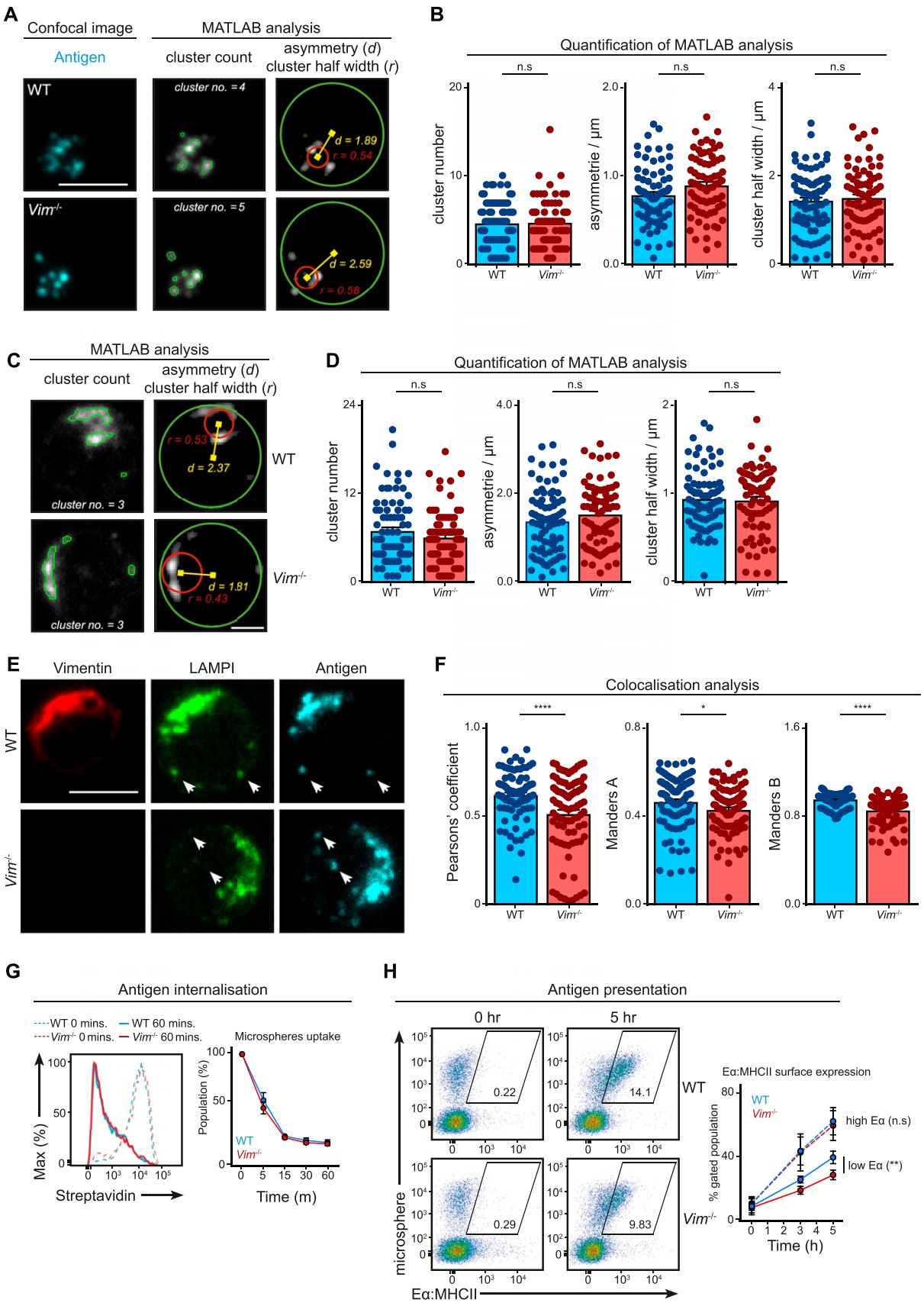

the antigen and LAMP1 using confocal microscopy (Fig 5A–D). To characterise polarisation of antigen and lysosome accurately, we analysed cropped images of single cells using MATLAB for cluster number, asymmetry, and cluster half-width per cell (Fig 5A–D) (see the Materials and Methods section). We found that WT and $Vim^{-/-}$ B cells exhibited similar cluster number, asymmetry, and cluster half-width of antigen-containing compartments (Fig 5A and B) and lysosomes (Fig 5C and D), indicating that the loss of vimentin did not impair the polarisation of both compartments in B cells. Notably, we found a modest but significant reduction in colocalisation of the two compartments (Figs 5E and F and S3F and G), suggesting that vimentin might play a subtle but specific role in the regulation of intracellular trafficking in B cells.

Next, we questioned whether this subtle difference in compartment colocalisation would affect subsequent antigen presentation; therefore, we went on to compare the ability to present antigen between WT and $Vim^{-/-}$ B cells using the E$\alpha$ presentation system (see the Materials and Methods section) (Fig 5G and H). Despite the fact that $Vim^{-/-}$ B cells exhibited no defects in microspheres uptake (Fig 5G), we observed a small but significant decrease in the presentation of E$\alpha$ peptide in $Vim^{-/-}$ B cells 3 and 5 h after incubation (Fig 5H), suggesting that these cells were indeed less effective in antigen presentation. Interestingly, this difference was abolished when E$\alpha$ was abundant (Fig 5H), suggesting that the role of vimentin might be more relevant in B cells that had acquired limited amount of antigen. Taken together, our data revealed a specific role of vimentin in the regulation of intracellular trafficking and antigen presentation.

## Vimentin is not required for B-cell differentiation in vitro

Given that vimentin might also play a role in B-cell activation in the longer term, we next asked whether loss of vimentin might influence other B-cell functions, such as proliferation and differentiation. To this end, we cultured CellTrace Violet (CTV)-labelled primary B cells from WT and $Vim^{-/-}$ mice with anti-IgM and IL-4 for 4 d in vitro (Fig 6A). We monitored cell proliferation of the cells by analysing the CTV dilution profile using flow cytometry (Fig 6A). We observed that both WT and $Vim^{-/-}$ B cells proliferated to a similar extent in response to anti-IgM stimulation (Fig 6A). We obtained similar results with LPS and anti-IgM/CD40L stimulation (Fig 6B). Taken together, these results suggested that vimentin was dispensable for activation-induced B-cell proliferation.

We went on to investigate if vimentin played a role in B-cell differentiation. Indeed, previous data had suggested that plasmablasts

lack vimentin (Dellagi et al, 1983). Using anti-IgM/CD40L stimulation of WT cells in vitro, we initially measured the level of vimentin in plasma cells and cells that had undergone a class switch recombination using flow cytometry (Fig S4A–C). We found that whereas IgG1 expression correlated with high level of vimentin, CD138 expression correlated with decreased level of vimentin (Fig S4A and B). We confirmed the decreased expression of vimentin in CD138$^+$ cells using confocal microscopy (Fig S4C). Moreover, we obtained similar comparisons in plasma cells generated in vivo through influenza infection (see the Materials and Methods section) in both the spleen and draining lymph node (Fig S4D and E). Taken together, our data suggested that vimentin down-regulation was a specific feature of plasma cell differentiation.

We thus asked whether the absence of vimentin influences B-cell differentiation in vitro. To this end, we cultured WT and $Vim^{-/-}$ B cells in LPS (Fig 6C) or anti-IgM/CD40L (Fig 6D) for 4 d and quantified both plasma cell differentiation and class switch recombination using flow cytometry (Fig 6C and D). However, we found no difference in the quantity of both CD138$^+$ cells and IgG1$^+$ cells between the culture of WT B cells and $Vim^{-/-}$ B cells in both culture conditions (Fig 6C and D), suggesting that vimentin was not required for either plasma cell differentiation or class switch recombination in vitro.

## Loss of vimentin delays antibody affinity maturation

Finally, we asked whether vimentin depletion in the B-cell compartment impaired humoral response in vivo. First, we carried out T cell–independent immunisation in WT and $Vim^{-/-}$ mice and characterised their IgM response after immunisation. To this end, we challenged mice with 2 $\mu$g NP-LPS via intraperitoneal injection and collected sera every 3 d following immunisation for antibody titre measurement by enzyme-linked immunosorbent assay (ELISA) (Fig 6E). We detected similar NP-specific IgM induction in both WT chimeras and $Vim^{-/-}$ chimeras following immunisation (Fig 6E). This was in line with our in vitro data showing that vimentin is not required for the activation and differentiation of B cells per se.

To address whether vimentin is required in a T cell–dependent immune response in a B-cell–intrinsic manner, we carried out immunisation and characterised immune response in mixed bone marrow chimeras. To this end, we transferred lethally irradiated $\mu$MT-KO mice with a bone marrow mixture containing 80% $\mu$MT-KO bone marrow and 20% WT or 20% $Vim^{-/-}$ bone marrow. The resulting chimera would harbour either WT or $Vim^{-/-}$ B cells in an environment of mostly WT cells. Next, we challenged successfully

**Figure 5. Vimentin regulates intracellular trafficking in B cells.**
**(A–F)** Primary B cells from WT and $Vim^{-/-}$ mice were stimulated using Alexa647-conjugated anti-IgM for 30 min. The cells were then fixed onto poly-lysine–coated coverslips and permeabilised. Next, the cells were labelled for LAMP1 suing anti-LAMP1 antibody and the corresponding secondary antibody. The samples were imaged using confocal microscopy. Single cells from the confocal images were then cropped and analysed individually using Matlab for cluster analysis. **(A, B)** The cluster number, asymmetry, and cluster half-width of antigen and (C, D) LAMP1 clusters were quantitated. Scale bar = 2 $\mu$m. **(E, F)** The degree of colocalisation between LAMP1 and antigen was analysed using Pearson's correlation and Manders coefficient. Scale bar = 5 $\mu$m. **(G)** Primary B cells from WT and $Vim^{-/-}$ mice were incubated with microspheres coated with biotinylated anti-IgM for 30 min on ice. Next, the cells were incubated at 37°C for up to 60 min. The cells were fixed at the end of the incubation and the remaining surface microspheres were detected using a fluorescent streptavidin. The samples were analysed using flow cytometry and the mean fluorescence intensity (MFI) of streptavidin was quantified. Data are representative of three independent experiments. **(H)** Primary B cells from WT and $Vim^{-/-}$ mice were incubated with microspheres coated with biotinylated anti-IgM and E$\alpha$ peptide for 30 min at 37°C. Next, the cells were incubated at 37°C for up to 5 h. The cells were fixed at the end of the incubation and the MHC-II–E$\alpha$ complexes were detected on the surface of the cells using a specific antibody and the corresponding secondary antibody. The samples were analysed using flow cytometry and the population of cells exhibiting high MFI (gated population) was quantified. Data are representative of three independent experiments.

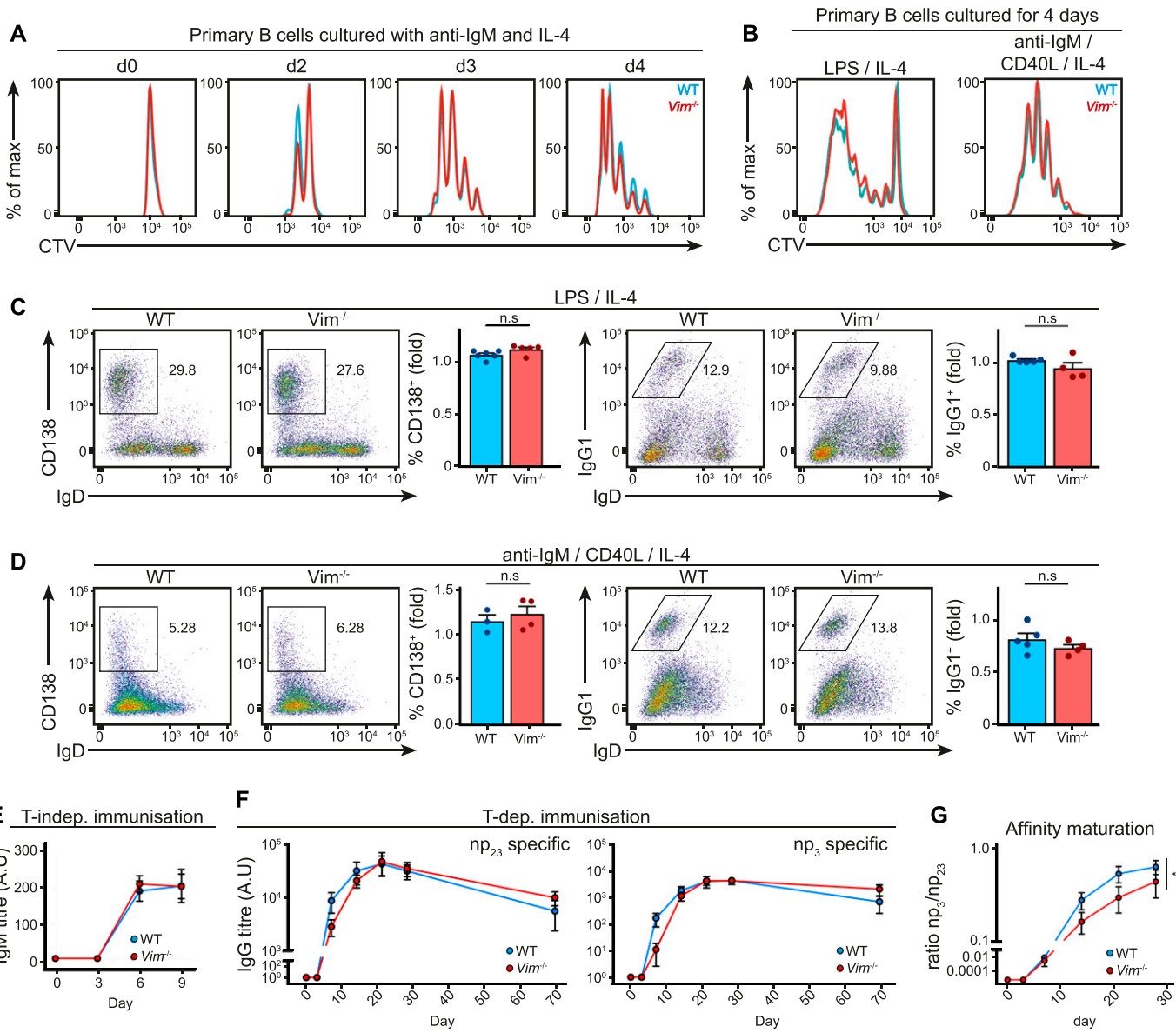

**Figure 6. Loss of vimentin in B cells impairs antibody response in vivo.**
**(A)** Representative histograms showing CTV dilution profiles of primary B cells cultured in vitro. Primary B cells from WT and $Vim^{-/-}$ mice were purified and cultured with anti-IgM and IL-4 in vitro for 4 d. The CTV dilution profile was monitored at different time points using flow cytometry to assess cell proliferation. **(B)** Representative histograms showing the CTV dilution profile of primary B cells cultured in vitro. Primary B cells from WT and $Vim^{-/-}$ mice were cultured for 4 d in the presence of LPS and IL-4 (left panel), or anti-IgM, CD40L, and IL-4 (right panel) in vitro. The CTV dilution was assessed on day 4 using flow cytometry. **(C, D)** Representative flow cytometry plots and the corresponding quantification showing B-cell differentiation into (left panel) plasma cells (CD138$^+$IgD$^-$) or (right panel) those that underwent class switch recombination (IgG1$^+$IgD$^-$) of B cells cultured in vitro for 4 d in the presence of (C) LPS and IL-4 or (D) anti-IgM, CD40L, and IL-4. **(E)** WT and $Vim^{-/-}$ mice were immunised with 2 µg NP-LPS via intraperitoneal injection and serum NP-specific IgM titre on days 3, 6, and 9 after immunisation was measured using ELISA. Two-way ANOVA was used to determine statistical significance. **(F)** WT and $Vim^{-/-}$ mixed bone marrow chimeras were immunised with 50 µg of NP conjugated to keyhole limpet haemocyanin with alum via intraperitoneal injections. Blood samples were collected on days −1, 3, 7, 14, 21, 28, and 69 after immunisation and serum antigen-specific IgG was measured using ELISA. Graphs depict the titres of anti-NP$_{23}$ (left panel) and anti-NP$_3$ (right panel) IgG. Two-way ANOVA was used to determine statistical significance. **(G)** The ratios of anti-NP$_{23}$ and anti-NP$_3$ IgG at each time point were calculated to determine the extent of affinity maturation of the IgG. Two-way ANOVA was used to determine statistical significance.

reconstituted chimeras with 50 µg NP conjugated to keyhole limpet haemocyanin plus alum and compared their antibody responses for up to 10 wk (Figs S4F and 6F and G). In WT chimeras, we detected a robust induction of both np$_{23}$-specific and np$_3$-specific IgG1 titres (Fig 6F). On the other hand, we detected a subtle delay in the generation of np$_{23}$-specific and np$_3$-specific IgG1 titre in the $Vim^{-/-}$ chimeras (Fig 6F), which together resulted in a modest but significant attenuation in the affinity maturation in the $Vim^{-/-}$ chimeras in response to immunisation (Fig 6G), suggesting that vimentin depletion in B cells affected the quality of antibody response in vivo.

# Discussion

In this study, we have evaluated in depth the role of IFs in B-cell activation. We have revealed that the collapse of cage-like vimentin distribution is rapidly induced by BCR signalling and is an enabling feature for morphology-altering processes immediately following B-cell activation. Indeed, inhibition of this collapse renders B cells irresponsive to antigen stimulation. On the other hand, genetic ablation of vimentin expression only had mild effects on antigen presentation but had no other apparent effects on B-cell development, activation, proliferation, and differentiation.

The observation of vimentin reorganisation upon BCR engagement was made previously, in which a capping response of vimentin was described (Dellagi et al, 1982). Here, we have extended the observations and revealed functional implications of this dramatic reorganisation in B-cell activation. The fact that this process is dependent on BCR signalling implicates it as a feature of BCR cross-linking and suggests a role in early B-cell activation. Here, we show that inhibition of this process severely impairs the responsiveness of B cells to antigen stimulation. As antigen-stimulated B cells undergo morphology-altering processes such as cell spreading (Fleire et al, 2006), we speculate that the collapse of the cage-like structure is favourable as it allows the membrane to become more flexible. Indeed, cage-like vimentin distribution provides more stiffness to the membrane of lymphocytes than collapsed vimentin distribution (Brown et al, 2001). Apart from deformability of the cell, our data also suggest that vimentin fluidity is likely required for intracellular antigen trafficking and polarisation, as WFA treatment completely abolishes antigen polarisation, possibly by inhibiting microtubule-mediated intracellular trafficking. This is consistent with the interacting characteristics between vimentin and microtubule (Chang & Goldman, 2004). We have shown that vimentin polarisation correlates with antigen polarisation, despite their mutual exclusivity.

Our study had revealed that the "basket" structure of vimentin filaments around internalised antigen probably plays a role in intracellular trafficking during B-cell activation. This is consistent with our observation that vimentin-deficient cells are less able to position intracellular compartments and present antigen in the context of low antigen availability. However, the mechanism in which vimentin controls intracellular trafficking remains to be elucidated. We reason that the subtle difference between WT and $Vim^{-/-}$ B cells in vitro might become more relevant in vivo, as cells will be in competition for scarcer antigen availability. In line with reduced antigen presentation in these cells, $Vim^{-/-}$ chimeras had defective affinity maturation of antigen-specific IgG1, which is indicative of as a less efficient germinal centre reaction.

Similar to previous studies (Dellagi et al, 1983), we demonstrated that in vitro–generated plasmablasts, but not class-switched cells, exhibit down-regulation of vimentin expression and collapse of the cage-like structure. Moreover, we show that activated B cells undergoing differentiation in vivo, both GC B cells and splenic plasmablasts, exhibit down-regulation of vimentin expression when compared with naive B cells. Consistent with our in vitro observations that dissolution and reorganisation of the vimentin network is necessary for the proper cellular functions in B cells immediately after antigen exposure, these results together suggest that a cage-like vimentin structure exerts a negative effect on activation-induced morphological changes such as cell blasting or cell division. In line with this, it was reported that bone marrow plasma cells express higher level of vimentin compared with plasmablasts (Jourdan et al, 2009). Despite the lack of major phenotype of $Vim^{-/-}$ B cells, the fact that WT mice contain vimentin-positive B cells strongly argue for a role of IFs in B cells from the evolutionary point of view.

In vivo, B cells regularly migrate and recirculate through both the blood and lymphatic system, during which they experience different level of shear stress. Cell–cell and cell–matrix interaction is also much more frequent in vivo. Indeed, vimentin has been shown to influence trans-endothelial migration of blood-derived lymphocytes (Nieminen et al, 2006). However, none of these are usually recapitulated in vitro. As the vimentin network significantly increases the stiffness of lymphocytes, it is tempting to extrapolate that $Vim^{-/-}$ B cells will experience different level of shear stress in vivo compared with WT B cells. This may in turn chronically alter the tonic signalling in $Vim^{-/-}$ B cells. Although it is not covered in the present study, it will certainly be interesting to compare the long-term fate of WT and $Vim^{-/-}$ B cells in vivo.

Overall, by using a combined approach of pharmacological and genetic dissection, we have established a fundamental role for the dynamic vimentin reorganisation in the immediate processes following antigen engagement to the BCR. We reason that the collapse vimentin network may increase the flexibility of the cells, which are needed for the rapid morphological changes in response to BCR signalling. We also uncovered a role of the vimentin cytoskeleton in regulating antigen presentation, details and mechanisms of which remain explored. Finally, our study provides many interesting observations that might shed light into future studies on the regulation of cytoskeleton in B cells.

# Materials and Methods

### Animal breeding and maintenance

Vimentin-deficient mouse was kindly provided by Balbino Alarcon, Universidad Autonoma de Madrid. Mice were backcrossed, bred, and maintained at the animal facility of the London Research Institute (the Francis Crick Institute). The Animal Ethics Committee of Cancer Research, UK; the Francis Crick Institute; and the UK Home Office approved all experiments.

### Immunisation and ELISA

For T-independent immunisation, mice were injected intraperitoneally with 2 $\mu$g of NP-LPS (Biosearch Technologies). Blood samples were taken from the lateral tail vein on days −1, 3, 6, and 9. For T-dependent immunisation, the mice were injected intraperitoneally with 50 $\mu$g NP$_{23}$-KLH (Biosearch Technologies) in 4 mg alum (Thermo Fisher Scientific). Blood samples were taken from the lateral tail vein on days −1, 3, 7, 14, 21, 28, and 69. In both cases, NP-specific antibody titres were detected by ELISA, using NP$_{23}$-BSA, NP$_3$-BSA (for T-dependent immunisation), and biotinylated anti-mouse IgM and

IgG (Southern Biotech). Titres were determined from the dilution curve in the linear range of absorbance. All noncommercial ELISA plates were developed with alkaline-phosphatase streptavidin (Sigma-Aldrich) and phosphorylated nitrophenyl-phosphate (Sigma-Aldrich). Absorbance at 405 nm was determined with a SPECTRAmas190 plate reader (Molecular Devices).

### Infection

For influenza infection, mice were intranasally immunised with 200 PFU of influenza A virus (PR8 strain). Draining mediastinal lymph node and spleen were taken on day 9 after virus administration.

### Cell isolation, labelling, and culture

Splenic naive B cells were purified using negative B-cell isolation kits (Miltenyi Biotec). Purified B cells were labelled in PBS with 2 $\mu$M CTV (Invitrogen) for 5 min at 37°C with 5% $CO_2$. The cells were maintained in complete B-cell medium: RPMI 1640 supplemented with 10% FCS, 25 mM Hepes, 1× Glutamax (Gibco), penicillin/streptomycin (Invitrogen), and 1% $\beta$-mercaptoethanol (Sigma-Aldrich).

### Proliferation analysis

CTV-labelled cells at a concentration of $10^6$ cells per mL were stimulated in complete B-cell medium supplemented with combinations of 1 $\mu$g/ml LPS (Sigma-Aldrich), 0.05 $\mu$g/ml CD40L (R&D Systems), 5 $\mu$g/ml anti-IgM F(ab)2 (Jackson ImmunoResearch), 10 ng/ml of IL-4 (R&D Systems), and 10 ng/ml IL-5 (R&D Systems). CTV dilution was measured after 4 d by flow cytometry.

### Antigen internalisation and presentation

For internalisation assays, purified B cells were loaded with biotinylated anti-IgM (SouthernBiotech) on ice for 20 min. Excess antibody was then removed, and the cells were incubated at 37°C for the time specified and fixed using 4% formaldehyde. Antibody remaining on the surface of the cells was labelled using fluorescent streptavidin (eBioscience) and analysed using flow cytometry. For antigen presentation assay, B cells were loaded with microspheres decorated by biotinylated anti-IgM (SouthernBiotech) and E$\alpha$ peptide for 30 min at 37°C. The cells were then washed to remove excess microspheres and were incubated at 37°C for the time specified. The cells were fixed using 4% formaldehyde and stained with anti-MHC-II:E$\alpha$ antibody (eBioY-Ae) (eBioscience), followed by anti-mouse IgG2b antibody (Life Technologies) staining for detection by flow cytometry. To measure microspheres uptake, the cells were incubated with coated microspheres for 30 min on ice. The cells were washed and incubated at 37°C for up to 1 h. The cells were fixed at specific time points and labelled with fluorescently labelled streptavidin and analysed using flow cytometry.

### Microsphere preparation

To prepare microspheres for presentation assays, 0.11 $\mu$m flash red microspheres (Bangs Laboratories) were incubated with a saturating amount of biotinylated anti-IgM (SouthernBiotech) and biotinylated E$\alpha$ peptide (generated by the peptide synthesis facility,

the Francis Crick Institute) for 1 h at 37°C. Microspheres were then washed twice with PBS to remove unbound complexes.

### Chemicals

Inhibitor against Syk BAY 61-3606 (Calbiochem), rapamycin (Sigma-Aldrich), and inhibitor against MEK PD184352 (Cell Signalling Technology) were used at a concentration of 1 $\mu$M, 100 nM, and 1 $\mu$M, respectively. WFA (Sigma-Aldrich) was used at a concentration of 10–20 $\mu$M. Lat A was used at a concentration of 1 $\mu$M. Alexa488-conjugated and Alexa647-conjugated phalloidin was obtained from Thermo Fisher Scientific.

### Flow cytometry

For analysis of lymphocyte populations, single-cell suspensions were prepared from homogenised spleens or bone marrow. For splenic suspension preparation, erythrocytes were destroyed with lysis buffer (BD Biosciences). The cells were treated with the appropriate combination of the following antibodies: CD16/32 (Fc block) (93) (eBioscience), B220 (RA3-6B2) (eBioscience), CD19 (eBio-1D3) (eBioscience), IgD (11-26c.2a) (eBioscience), IgM (II/41) (BioLegend), CD43 (S7) (BioLegend), CD24 (M1/69) (BD Biosciences), CD21 (7G6) (BD Biosciences), and BP-1/Ly-51 (6C3) (BioLegend). To identify splenic plasma cells or plasma cells from lymph node in vivo, cells positive for CD138 (281.2) (eBioscience) and negative for IgD (11-26c.2a) were considered plasma cells. For analysis of in vitro B-cell cultures, after blocking Fc receptors using anti-CD16/32 antibody, CTV-labelled cells were stained with the antibodies CD138 (281.2) and IgG1 (A85.1). For detection of vimentin using flow cytometry, anti-vimentin antibody (ERP3776) (Abcam) was used.

### Immunoblotting

Purified B cells were left at 37°C for at least 10 min in the medium to equilibrate before stimulation. They were then stimulated for various times with 5 $\mu$g/ml anti-IgM F(ab)2 fragment (Jackson ImmunoResearch). For immunoblotting, stimulated cells were then lysed in lysis buffer (20 mM Tris–HCL, pH 8.0, 150 mM NaCl, 5 mM EDTA, protease inhibitor cocktail [Roche], 10 mM NaF, 1 mM Na3VO4, and 1% NP40) for 30 min on ice. The samples were loaded onto 12% PAGE gel (BioRad) for electrophoresis. Antibodies against p-CD19 (Tyr 531), p-Syk (Tyr 352), p-Akt (Ser 473), and p-Erk (Thr 202/Tyr 204) (Cell Signalling Technology), and secondary HRP-conjugated anti-rabbit antibody were used to probe B-cell signalling. Blot densitometry analysis was performed using ImageJ (National Institutes of Health) software.

### High-throughput imaging and analysis

Cells were settled onto a poly-lysine–coated 96-well flat-bottom microplate (Falcon). The cells were fixed using 2% paraformaldehyde and processed for microscopy. The samples were imaged using the Cellomics ArrayScan VTI Live Cell Module (Thermo Fisher Scientific) system. The images were processed and quantitated using the HCS navigator system (Thermo Fisher Scientific) using the spot identification module in the software.

## Sample preparation for microscopy

Cells were settled onto poly-lysine–coated 35-mm dishes (MatTek Corporation) for 5 min at 37°C. The samples were fixed using 2% paraforamdehyde for at least 40 min and permeabilised using 0.1% Triton for 1 min. The samples were incubated with appropriate primary and secondary antibodies. Antibodies used for immunofluorescence: anti-vimentin (ERP3776) (Abcam), anti-$\alpha$-tubulin (B-5-1-2) (Sigma-Aldrich), anti-LAMP1 (1D4B) (BD Biosciences), goat–anti-rabbit IgG Alexa488, goat–anti-rabbit IgG Alexa647, goat–anti-mouse IgG1 Alexa555, and goat-anti-rat IgG Alexa488 (Thermo Fisher Scientific).

## Optical microscopy

Confocal imaging was performed with an LSM 780 microscope (Carl Zeiss) with a plan apochromat 63×, NA 1.40 objective. The images were analysed with Imaris (Bitplane) or ImageJ software.

## Airyscan confocal microscopy

Airyscan was performed with an LSM 880 microscope (Carl Zeiss) with a plan apochromat 63×, NA 1.40 objective. The images were processed using Zen Black software (Carl Zeiss) and analysed with ImageJ (NIH).

## SIM

SIM was performed on an Elyra PS.1 microscope (Carl Zeiss) using 488, 561, and 640 nm laser excitation and a 63×/1.40 plan apochromat oil-immersion objective (Carl Zeiss). Two-colour alignment was performed using a multi-colour bead sample (Carl Zeiss) and the channel alignment function in the Zen software (Carl Zeiss). The images were reconstructed in the Zen software using a theoretical point-spread function and a noise filter setting of −4.0 for both channels. SIM images were corrected for shift between the different channels using images of TetraSpeck beads (Thermo Fisher Scientific) in Zen software (Carl Zeiss).

## TIRFM/dSTORM

The cells were fixed with 2% paraformaldehyde and labelled as described above. For imaging, the samples were put in freshly prepared Tris buffer (pH 8.0) containing 15 mM cystamine (Sigma-Aldrich) that was complemented with colloidal gold particles (100 nm diameter, British Biocell) as fiducials. The samples were imaged using LSM 780/Elyra PS.1 microscope (Carl Zeiss).

## Cluster and colocalisation analysis

Cluster analysis, asymmetry, and spatial distribution of clusters were analysed in Matlab (The MathWorks Inc.). Maximum intensity projections of z-stacks were bandpass-filtered with a filter size between 0.13 and 0.9 $\mu$m, segmented using the method by Otsu, and the number of the segmented clusters recorded for each cell (Otsu, 1979). The centre of the intensity distribution in the clustered channel was calculated as the mean pixel position weighted by the pixel intensity after background subtraction with a 2.6-$\mu$m radius. The mean for x and y were calculated independently and then averaged. The centre of the cell was determined from the image of the cell nucleus by manually overlaying the image with a circular mask. Asymmetry was calculated as the distance between the mean of the intensity distribution (as described above) and the centre of the cell. The width of the spatial distribution of the clusters was determined by calculating the standard deviation of the pixel positions weighted with pixel intensity of the clusters, again separately in x and y and then averaged. To quantify colocalisation, the images were cropped to the size of the cell, a threshold was applied using the method by Costes et al (2004), and the Pearson's coefficient was calculated using Imaris software (Bitplane) (Costes et al, 2004). Statistical significance was assessed using $t$ test (two tailed, $P < 0.05$) using Prism software (GraphPad).

## Experimental data and statistical analysis

The sample sizes were chosen on the basis of published work in which similar phenotypical characterisation and similar defects were reported. Cohort randomisation or "blinding" of investigators to sample identity was not carried out in this study. For all statistical comparisons unless specified, the data for each group were compared with $t$ test and $P$-values were calculated. Normal distribution of samples was assumed on the basis of published studies with analyses similar to ours. Statistically significant differences are indicated on the figures as follows: *$P < 0.05$, **$P < 0.005$, ***$P < 0.0005$, and ****$P < 0.00005$.

# Supplementary Information

# Acknowledgements

We thank Michael Howell of the High Throughput unit (CRUK and the Francis Crick Institute) for his assistance. We thank the Biological Resource Unit (CRUK and the Francis Crick Institute) for animal maintenance and the flow cytometry unit (CRUK and the Francis Crick Institute) for technical support. We thank all members of the Lymphocyte Interaction Laboratory (the Francis Crick Institute and Ragon Institute) for discussions and comments. All the work presented here was supported by the Francis Crick Institute core funded by Cancer Research UK (FC001035 and FC001136), the UK Medical Research Council (FC001035 and FC001136), and the Wellcome Trust (FC001035 and FC001136); UCL ORS awards to C Tsui, a Marie Skodowska-Curie individual postdoctoral fellowship to N Martinez-Martin, the Center for HIV/AIDS Vaccine Immunology and Immunogen Discovery of the National Institutes of Health (UM1AI100663), the Philip T. and Susan M. Ragon Institute Foundation, and an NIH R01 grant (AI135052-01A1) to FD Batista.

## Author Contributions

C Tsui: investigation
N Martinez-Martin: supervision
P Maldonado: investigation
B Montaner: resources

A Borroto: resources
B Alarcon: resources
A Bruckbauer: methodology
FD Batista: supervision

**Conflict of Interest Statement**

The authors declare that they have no conflict of interest.

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
