## [Reviewer comments · Life Science Alliance]

Life Science Alliance

Dynamic reorganisation of intermediate filaments coordinates early B cell activation

Carlson Tsui, Nuria Martinez-Martin, Paula Maldonado, Beatriz Montaner, Aldo Borroto, Balbino Alarcon, Andreas Bruckbauer, and Facundo Batista

Corresponding author(s): Facundo Batista, Ragon Institute

Review Timeline:

Submission Date:	2018-03-27
Editorial Decision:	2018-05-28
Revision Received:	2018-07-02
Editorial Decision:	2018-08-06
Revision Received:	2018-08-08
Accepted:	2018-08-09

Scientific Editor: Andrea Leibfried

Transaction Report:

May 28, 2018

Re: Life Science Alliance manuscript #LSA-2018-00060

Dr. Facundo D. Batista
Ragon Institute
400 Technology Square
Cambridge, MA 02139-3583

Dear Dr. Batista,

Thank you for submitting your manuscript entitled "Dynamic reorganisation of intermediate filaments coordinates early B cell activation" to Life Science Alliance. The manuscript was assessed by expert reviewers, whose comments are appended to this letter. Please excuse the delay in getting back to you with a decision, the reviewing of your manuscript took longer than anticipated.

As you can see below, the manuscript got split reviews. While reviewer #1 and #3 voice some technical concerns but leave room for further consideration, reviewer #2 is against publication, and this reviewer stressed this view again once all reviews were received. This vimentin expert thinks that the results using vimentin null cells negate the role attributed to vimentin in vitro, and in the reviewer's opinion the study is therefore not suitable for publication. However, reviewer #1 and #3 support further consideration of a revised version. You already provided an outline to us describing how you would address the criticisms raised, especially those regarding vimentin organization, missing proof of a direct effect of WFA treatment on vimentin, as well as the criticism regarding the sample size. We discussed your outline and we decided to invite you to submit a revision if you can address the reviewers' key concerns as proposed.

-- High-resolution figure, supplementary figure and video files uploaded as individual files: See our detailed guidelines for preparing your production-ready images, <http://life-science-alliance.org/authorguide>

B. MANUSCRIPT ORGANIZATION AND FORMATTING:

Full guidelines are available on our Instructions for Authors page, <http://life-science-alliance.org/authorguide>

Thank you for this interesting contribution to Life Science Alliance. We are looking forward to receiving your revised manuscript.

Sincerely,

Reviewer #1 (Comments to the Authors (Required)):

The paper explores the role of vimentin in B cell function using super-resolution microscopy, pharmacological and knockout mouse studies. Prior studies had shown that vimentin forms a cage in circulating human lymphocytes that makes the less deformable and may protect them in

circulation. Activation of cells with chemokines causes collapse of the vimentin to the uropod of polarized cells. They recapitulate these findings to some extent with murine splenic B cells, but they only look at antigen activation with BCR crosslinking or surface immobilized anti-BCR. The results are interesting and suggest that stabilizing vimentin with a drug prevents the reorganization and leads to an atypical localization of antigen containing endosomes and localization with Lamp-1 positive compartments. The knockout mice show a milder defect organization of internalized antigen, but they have a defect in germinal center function that manifests as slower production of high affinity antibodies in vimentin deficient cells, suggesting defects in antigen presentation.

Major issues:

The original studies on vimentin cages were with peripheral blood cell, where the cage might be useful to resist shear forces and then retract to uropod on activation by chemokines. So I'm surprised that splenic B cells, which are mostly in B cell zones, would have the complete cage. The authors should comment on this. Would it be possible to do some perfusion fixation and histology to determine if follicular B cells have a vimentin cage in polarized B cells in follicles or if its polarized, but loses polarization during process of isolation in vitro. This seems important to think about the biology of this. Of course the idea about the importance of the cage in flow was somewhat speculative from the start and is not supported by the apparent health of all B cell population in the *vim*^{-/-} mice.

The authors use Withaferin A, which has some effect on intermediate filament rearrangement, but otherwise there seems to be no idea how this works or what other systems it impacts. Since intermediate filaments have no known motors and don't undergo a polarized filament treadmill like F-actin there is no way for it to "move" so its failure to reorient probably reflects a problem with F-actin or microtubules. The effect of Withaferin A on these other system should be tested. The effects on spreading suggest a problem with F-actin, but they should look at the other systems directly to leave no doubt. It would also be helpful to use Withaferin A on the *Vim*^{-/-} cells introduced later in the paper to confirm that Withaferin A effects require vimentin.

The characterization of the *vim*^{-/-} mice is well done. The delay in making high affinity anti-NP antibodies at d7 is clear in 6F. However, its not obvious how 6F and 6G fit together. The production of the high affinity anti-NP seems to catch up by d21, but this is the day on which the ratio of NP3/NP21 shows that largest difference. Also this is a linear scale, where data in 6F suggests that this difference should require a log scale to plot as the ratio of high to low at day 7 should be 0.01-0.001 (by eye). The authors should check this.

Minor issues

When they use the term antigen experienced on page 6 this could be misconstrued at memory rather than just recently activated by BCR cross-linking. In context its understandable, but should probably restate it just to avoid confusion nonetheless.

The authors refer to antigen presentation in Figure 5G, this implies interaction with a T cell, which is not incorporated. The assay has the potential to determine if the defect is related to microbead uptake, processing or both by analyzing the total microbead positive B cells and the fraction of these that express the Eα peptide. It looks to me like the defect may be in microbead uptake, rather than in endocytic generation of pMHC.

Reviewer #2 (Comments to the Authors (Required)):

The paper by Tsui, et al entitled "Dynamic Reorganization of Intermediate Filaments Coordinates Early B-cell Activation" attempts to demonstrate a "fundamental role" for the reorganization of vimentin rapidly following exposure to agents that stimulate the B-cell receptor response. They conclude that the changes in vimentin are involved in regulating the presentation of antigens on the cell surface and the resulting signaling pathways. They also state that a "rigid intermediate filament cytoskeleton" inhibits cells spreading and the trafficking of antigens within the cytoplasm.

Overall this paper uses mainly microscopic methods to draw conclusions regarding the role of vimentin in the presentation of antigen and the internal movements or trafficking of antigens. These conclusions are drawn mainly from imaging with confocal microscopy and the super-resolution methods afforded by STORM and SIM. Their major observation for these various imaging techniques is related to the "collapse" of the extended intermediate filament system to "one pole" of the cell which is correlated with the redistribution of surface antigen. This collapsed structure contains some filamentous elements at their edges, but similar structures were not seen in the center of the cluster where the antigen seemed to reside. From their STORM and SIM image analysis they conclude that there are no changes in the structure or thickness of the filaments. The images presented are not convincing and they do not demonstrate that they have the resolution required to see structural changes in intermediate filaments (~ 10nm in diameter) using light microscopy. They also state that the so called collapse of vimentin is accompanied by a complete loss of vimentin in the peripheral regions of the cell. However there are numerous reports that vimentin exists in several assembly states in cells including particles, short filaments and unit length filaments which could be present in what the authors define as "lamellae" at the cell periphery. Also they state that the central region of the "vimentin clump" is dense and unresolvable. This clump is likely to contain large numbers of densely packed vimentin filaments and if their light microscopy procedures cannot document this then they should use electron microscopy.

The authors use WFA to induce vimentin bundling followed by BCR and state that the cells failed to reorganize vimentin. This is very confusing since WFA appears to cause the same type of aggregation as B cell activation. They also suggest that WFA cross links vimentin. There is no evidence that WFA can cross link vimentin in this paper nor in the literature. Furthermore there is significant evidence that other elements of the cytoskeleton, such as microtubules and actin are altered when intermediate filaments are disrupted. No mention is made of these findings which certainly could alter their conclusions.

Finally the authors did the most important set of experiments by using primary B lymphocytes from WT and vimentin null mice. However, they failed to detect any significant differences in the developmental pathway of B cells in either the spleen or bone marrow in the null mice. In addition, they showed no differences in BCR signaling, internalization of antigen, etc. They state that "...our data indicated that the lack of vimentin does not hinder morphologic-altering processes during B-cell activation." This seems contradictory and appears to negate many of their conclusions.

Reviewer #3 (Comments to the Authors (Required)):

1. I will begin by saying that I truly enjoyed reading this text as the story is easy to follow and it is possible to understand how the authors went from one experiment to the next. In other words, it is possible to see that logic - the "red thread" - through the entire text. I also found that the results were truly intriguing and fascinating in matter of vimentin reorganization in the early B-cell

activation and I feel that this is a good step forward in the understanding of the importance of intermediate filaments in B-cell activation. The imaging results are in general very beautiful to look at, but I must say that I was a bit disappointed that the authors haven't used a stronger microscope for the imaging experiments, as a lot of the resolution is lost with normal confocal, especially when it comes to the zoomed in pictures.

2. As a lot of data has already been obtained in the current version of the article, I feel that it is not necessary to begin any new experiments. I would however like to see that the authors do at least 3 independent experiments, as currently they have stated in a number of figure texts that they have done "at least 2 independent experiments". Doing at least 3 experiments would give the results more credibility, from a statistical point of view. This would probably take a bit of time considering the amount of obtained data, but I also feel that this is necessary.

3. I had a few issues with the text. The authors have for example described the procedure of the experiment in a lot of detail, with time, temperature and so on in the actual text. I feel this should either be kept in the materials and methods section, or moved to the corresponding figure texts, where the description should be shortened and very concise, as the focus in the section are the results, not the procedure.

The authors have also exaggerated in a bit in a few places by saying for example that "they employed "the powerful" microscopy method...". It is of course good that a method is powerful, but there is no need to advertise it in an article.

Finally, I would also like to see the authors patch up their references a bit. Currently the number of the journal is so much bigger than the rest of the reference, that it starts to look really odd. If they want to highlight the journal number, they should consider using bold lettering instead of bigger font size. I also noticed that there is a mix of formats in the journal names, as some references have the full journal name and other have the abbreviated version. An example of this is that the authors have written "Nat Rev Immunol" in one reference and "The Journal of Immunology" in another. Be consistent.

Point-by-point Outline - Life Science Alliance: LSA-2018-00060

(Reviewer #1)

Major issues:

The original studies on vimentin cages were with peripheral blood cell, where the cage might be useful to resist shear forces and then retract to uropod on activation by chemokines. So I'm surprised that splenic B cells, which are mostly in B cell zones, would have the complete cage. The authors should comment on this. Would it be possible to do some perfusion fixation and histology to determine if follicular B cells have a vimentin cage in polarized B cells in follicles or if its polarized, but loses polarization during process of isolation *in vitro*? This seems important to think about the biology of this. Of course, the idea about the importance of the cage in flow was somewhat speculative from the start and is not supported by the apparent health of all B cell population in the *vim*^{-/-} mice.

Our Response: We agree with **R1** on this issue. To address this, we have performed Airyscan super-resolution confocal microscopy to inspect frozen splenic sections of WT mice. We have observed that cage-like vimentin distribution was also present in B cells *in vivo*. We hope will be sufficient to clarify this, and we have included the results in the modified manuscript (**Supplementary Figure 1B**).

The authors use Withaferin A, which has some effect on intermediate filament rearrangement, but otherwise there seems to be no idea how this works or what other systems it impacts. Since intermediate filaments have no known motors and don't undergo a polarized filament treadmill like F-actin there is no way for it to "move" so it's its failure to reorient probably reflects a problem with F-actin or microtubules. The effect of Withaferin A on these other systems should be tested. The effects on spreading suggest a problem with F-actin, but they should look at the other systems directly to leave no doubt. It would also be helpful to use Withaferin A on the *Vim*^{-/-} cells introduced later in the paper to confirm that Withaferin A effects require vimentin.

Our response: **R1** raised an important point with regards to possible secondary effects of WFA on actin and microtubules. This raised the possibility of cooperation between actin, microtubule and vimentin. To formally establish this, we have performed new experiments not only with well characterised inhibitors of actin and microtubule, but also genetic experiments to prove this link. Indeed, we found that Latrunculin A or nocodazole interferes with the reorganisation of vimentin. Furthermore, genetic ablation of CDC42, a key regulator of actin cytoskeleton and B cell polarity, also inhibited vimentin reorganisation. These new experimental data support the notion of a coordinating action among actin, microtubule and vimentin (**Figure 4**). Lastly, to simplify the message, we have decided to take off many of the experiments regarding to the use of WFA. We would like to thank **R1** for raising the point, which we believe have substantially improved our message.

The characterization of the *vim*^{-/-} mice is well done. The delay in making high affinity anti-NP antibodies at d7 is clear in 6F. However, it's not obvious how 6F and 6G fit together. The production of the high affinity anti-NP seems to catch up by d21, but this is the day on which the ratio of NP3/NP21 shows that largest difference. Also, this is a linear scale, where data in 6F suggests that this difference should require a log scale to plot as the ratio of high to low at day 7 should be 0.01-0.001 (by eye). The authors should check this.

Our response: We thank **R1** for this feedback and the specific suggestion. To further characterise the nature of the titre differences, we have extended our antibody titre characterisation of immunised mixed bone-marrow chimeras to 10 weeks after immunisation, and also re-formatted the display of affinity maturation. These are now included in the revised manuscript (**Figure 6F and 6G**). We hope these will better translate our observations.

Minor issues

When they use the term antigen experienced on page 6 this could be misconstrued at memory rather than just recently activated by BCR cross-linking. In context it's understandable, but should probably restate it just to avoid confusion nonetheless.

Our response: We thank **R1** for this suggestion and we apologise for the choice of words. To address this, "Antigen-experienced" is now replaced by "activated".

The authors refer to antigen presentation in Figure 5G, this implies interaction with a T cell, which is not incorporated. The assay has the potential to determine if the defect is related to microbead uptake, processing or both by analyzing the total microbead positive B cells and the fraction of these that express the Ealpha peptide. It looks to me like the defect may be in microbead uptake, rather than in endocytic generation of pMHC.

Our response: We completely agree with **R1** for this important issue. We have now included data showing the uptake kinetics of microspheres of *Vim*-ko cells in the revised manuscript (**Figure 5G**).

(Reviewer #2)

Comments to the Authors (Required) The paper by Tsui, et al entitled "Dynamic Reorganization of Intermediate Filaments Coordinates Early B-cell Activation" attempts to demonstrate a "fundamental role" for the reorganization of vimentin rapidly following exposure to agents that stimulate the B-cell receptor response. They conclude that the changes in vimentin are involved in regulating the presentation of antigens on the cell surface and the resulting signaling pathways. They also state that a "rigid intermediate filament cytoskeleton" inhibits cells spreading and the trafficking of antigens within the cytoplasm. Overall this paper uses mainly microscopic methods to draw conclusions regarding the role of vimentin in the presentation of antigen and the internal movements or trafficking of antigens. These conclusions are drawn mainly from imaging with confocal microscopy and the super-resolution methods afforded by STORM and SIM. Their major observation for these various imaging techniques is related to the "collapse" of the extended intermediate filament system to "one pole" of the cell which is correlated with the redistribution of surface antigen. This collapsed structure contains some filamentous elements at their edges, but similar structures were not seen in the center of the cluster where the antigen seemed to reside. From their STORM and SIM image analysis they conclude that there are no changes in the structure or thickness of the filaments. The images presented are not convincing and they do not demonstrate that they have the resolution required to see structural changes in intermediate filaments (~ 10nm in diameter) using light microscopy.

Our response: We did not wish to claim that our microscopy method would detect structural changes below the resolution limit of the technique, and we apologise for any misinterpretation of claims our original manuscript might have caused. We have now reworded our comment regarding the structure of vimentin in the revised manuscript.

They also state that the so-called collapse of vimentin is accompanied by a complete loss of vimentin in the peripheral regions of the cell. However there are numerous reports that vimentin exists in several assembly states in cells including particles, short filaments and unit length filaments which could be present in what the authors define as "lamellae" at the cell periphery.

Our response: We have now modified our wording regarding the lack of mature vimentin filaments in the periphery in the revised manuscript. Similar to the last point, we do not rule out the existence of vimentin particles and short filaments in the periphery that might fall below the resolution limits.

Also they state that the central region of the "vimentin clump" is dense and unresolvable. This clump is likely to contain large numbers of densely packed vimentin filaments and if their light microscopy procedures cannot document this then they should use electron microscopy. The authors use WFA to induce vimentin bundling followed by BCR and state that the cells failed to reorganize vimentin. This is very confusing since WFA appears to cause the same type of aggregation as B cell activation. They also suggest that WFA cross links vimentin. There is no evidence that WFA can cross link vimentin in this paper nor in the literature.

Our response: We apologise for the poor choice of word when we assigned the role of WFA on vimentin. We have now clarified the documented action of WFA on vimentin in the revised manuscript. We hope the changes will be satisfactory for **R2**.

Furthermore, there is significant evidence that other elements of the cytoskeleton, such as microtubules and actin are altered when intermediate filaments are disrupted. No mention is made of these findings which certainly could alter their conclusions.

Our response: We thank **R2** for raising this important point which share some similarity to that of **R1**. As laid out in the revised manuscript, we have now reconsidered and performed new experiments to address the possible cooperativity of actin and microtubules on vimentin dynamics (**Figure 4**). We have also commented on the established cross-talk between the cytoskeleton systems.

Finally, the authors did the most important set of experiments by using primary B lymphocytes from WT and vimentin null mice. However, they failed to detect any significant differences in the developmental pathway of B cells in either the spleen or bone marrow in the null mice. In addition, they showed no differences in BCR signaling, internalization of antigen, etc. They state that "...our data indicated that the lack of vimentin does not hinder morphologic-altering processes during B-cell activation." This seems contradictory and appears to negate many of their conclusions.

Our response: From the previous modifications suggested by both **R1** and **R2**, we hope our revised message is now improved and non-contradictory, and would be satisfactory for **R2**.

(Reviewer #3)

Comments to the Authors (Required) 1. I will begin by saying that I truly enjoyed reading this text as the story is easy to follow and it is possible to understand how the authors went from one experiment to the next. In other words, it is possible to see that logic - the "red thread" - through the entire text. I also found that the results were truly intriguing and fascinating in matter of vimentin reorganization in the early B-cell activation and I feel that this is a good step forward in the understanding of the importance of intermediate filaments in B-cell activation. The imaging results are in general very beautiful to look at, but I must say that I was a bit disappointed that the authors haven't used a stronger microscope for the imaging experiments, as a lot of the resolution is lost with normal confocal, especially when it comes to the zoomed in pictures.

Our response: We thank **R3** for the generous remarks. Regarding the comment on imaging techniques, we have performed new experiments which included super resolution Airyscan technique to reinforce our original conclusions, including vimentin reorganisation, antigen "pockets" and antigen-Lamp1 colocalisation. These are now included in the revised manuscript (**Figure 2D, Figure 4C-4D, Supplementary Figure 1A, Supplementary Figure 3F**).

2. As a lot of data has already been obtained in the current version of the article, I feel that it is not necessary to begin any new experiments. I would however like to see that the authors do at least 3 independent experiments, as currently they have stated in a number of figure texts that they have done "at least 2 independent experiments". Doing at least 3 experiments would give the results more credibility, from a statistical point of view. This would probably take a bit of time considering the amount of obtained data, but I also feel that this is necessary.

Our response: We have performed new experiments and analysed more previously experiments to ensure that our data in our revised manuscript is representative of at least 3 independent experiments.

3. I had a few issues with the text. The authors have for example described the procedure of the experiment in a lot of detail, with time, temperature and so on in the actual text. I feel this should either be kept in the materials and methods section, or moved to the corresponding figure texts, where the description should be shortened and very concise, as the focus in the section are the results, not the procedure.

The authors have also exaggerated in a bit in a few places by saying for example that "they employed "the powerful" microscopy method...". It is of course good that a method is powerful, but there is no need to advertise it in an article. Finally, I would also like to see the authors patch up their references a bit. Currently the number of the journal is so much bigger than the rest of the reference, that it starts to look really odd. If they want to highlight the journal number, they should consider using bold lettering instead of bigger font size. I also noticed that there is a mix of formats in the journal names, as some references have the full journal name and other have the abbreviated version. An example of this is that the authors have written "Nat Rev Immunol" in one reference and "The Journal of Immunology" in another. Be consistent.

Our response: We apologise for the poor layout and formatting in some of the areas of the original manuscript. We have also removed words such as "powerful" that could be seen as exaggeration. Finally, we have re-formatted our reference list. We hope these will be satisfactory for both **R3** and the standard of *LSA*.

August 6, 2018

RE: Life Science Alliance Manuscript #LSA-2018-00060R

Dr. Facundo D. Batista
Ragon Institute
400 Technology Square
Cambridge, MA 02139-3583

Dear Dr. Batista,

Thank you for submitting your revised manuscript entitled "Dynamic reorganisation of intermediate filaments coordinates early B cell activation". I apologize for the delay in getting back to you, we were waiting for further input on your work that was not provided in the end. However, as you can see in the attached report below, reviewer #1 is satisfied with the revision performed and now supports publication. We also appreciate your response to the concerns raised by reviewer #2 and #3, and we would be happy to publish your paper in Life Science Alliance pending final revisions necessary to meet our formatting guidelines. Congratulations on this nice work!

Please upload a final version of your manuscript, paying attention to the following:

- please provide a running title and a summary blurb during submission, both will be displayed on the landing page of Life Science Alliance, so they should be complementary to the title and informative
- please change your reference list to 10 authors et al. (currently 6 authors et al. are listed)
- please add scale bars to the following figure panels: Fig2E, Fig4C, E, G, H, I, J, Fig5C, FigS1A and B, FigS3B, C, D, F, FigS4C
- please indicate in Fig4A the origin of magnification of vimentin

You will be guided to complete the submission of your revised manuscript and to fill in all necessary information, including the license to publish form.

A. FINAL FILES:

-- High-resolution figure, supplementary figure and video files uploaded as individual files: See our detailed guidelines for preparing your production-ready images, <http://life-science-alliance.org/authorguide>

-- Summary blurb (enter in submission system): A short text summarizing in a single sentence the study (max. 200 characters including spaces). This text is used in conjunction with the titles of

papers, hence should be informative and complementary to the title. It should describe the context and significance of the findings for a general readership; it should be written in the present tense and refer to the work in the third person. Author names should not be mentioned.

B. MANUSCRIPT ORGANIZATION AND FORMATTING:

Full guidelines are available on our Instructions for Authors page, <http://life-science-alliance.org/authorguide>

Sincerely,

Andrea Leibfried, PhD
Executive Editor
Life Science Alliance
Meyershofstr. 1
69117 Heidelberg, Germany
t +49 6221 8891 502
e a.leibfried@life-science-alliance.org
www.life-science-alliance.org

Reviewer #1 (Comments to the Authors (Required)):

The authors have address my concerns. The new results clarify the nature of vimentin organisation in B cells in the steady state in lymphoid tissues and the effect of activation. The new experiments with WFA and the focusing of its use improves both understanding of its effects and hones the message of the paper. The authors have addressed all my minor concerns. Very little has been done on the role of vimentin in immune cells and thus the work is novel and timely.

August 9, 2018

RE: Life Science Alliance Manuscript #LSA-2018-00060RR

Dr. Facundo D. Batista
Ragon Institute
400 Technology Square
Cambridge, MA 02139-3583

Dear Dr. Batista,

Thank you for submitting your Research Article entitled "Dynamic reorganisation of intermediate filaments coordinates early B cell activation". It is a pleasure to let you know that your manuscript is now accepted for publication in Life Science Alliance. Congratulations on this interesting work.

The final published version of your manuscript will be deposited by us to PubMed Central (PMC) as soon as we are allowed to do so, the application for PMC indexing has been filed. You may be eligible to also deposit your Life Science Alliance article in PMC or PMC Europe yourself, which will then allow others to find out about your work by Pubmed searches right away. Such author-initiated deposition is possible/mandated for work funded by eg NIH, HHMI, ERC, MRC, Cancer Research UK, Telethon, EMBL.

Please also see:

<https://www.ncbi.nlm.nih.gov/pmc/about/authorms/>

<https://europepmc.org/Help#howsubsmanu>

DISTRIBUTION OF MATERIALS:

Again, congratulations on a very nice paper. I hope you found the review process to be constructive and are pleased with how the manuscript was handled editorially. We look forward to future exciting submissions from your lab.

Sincerely,
